# Evaluating Large Multimodal Models for Nutrition Analysis: A New Benchmark Enriched with Contextual Metadata

Bruce Coburn*, Jiangpeng He†§, Megan E. Rollo‡, Satvinder S. Dhaliwal‡, Deborah A. Kerr‡, Fengqing Zhu*

*Abstract*—Large Multimodal Models (LMMs) are increasingly applied to meal images for nutrition analysis. However, existing work primarily evaluates proprietary models, such as GPT-4. This leaves the broad range of LMMs underexplored. Additionally, the influence of integrating contextual metadata and its interaction with various reasoning modifiers remains largely uncharted. This work investigates how interpreting contextual metadata derived from GPS coordinates (converted to location/venue type), timestamps (transformed into meal/day type), and the food items present can enhance LMM performance in estimating key nutritional values. These values include calories, macronutrients (protein, carbohydrates, fat), and portion sizes. We also introduce ACETADA, a new food-image dataset. This public dataset provides nutrition information verified by the dietitian and serves as the foundation for our analysis. Our evaluation across eight LMMs (four open-weight and four closed-weight) first establishes the benefit of contextual metadata integration over straightforward prompting with images alone. We then demonstrate how this incorporation of contextual information enhances the efficacy of reasoning modifiers, such as Chain-of-Thought, Multimodal Chain-of-Thought, Scale Hint, Few-Shot, and Expert Persona. Empirical results show that integrating metadata intelligently, when applied through straightforward prompting strategies, can significantly reduce the Mean Absolute Error (MAE) and Mean Absolute Percentage Error (MAPE) in predicted nutritional values. This work highlights the potential of context-aware LMMs for improved nutrition analysis.

*Index Terms*—Large Multimodal Model, Nutrition Analysis, Portion Estimation, Prompt Engineering.

## I. INTRODUCTION

Image-based nutrition analysis offers a practical and automated alternative to self-report methods such as weighed food records, 24-h recalls (ASA24), and manual logging apps (e.g., MyFitnessPal) ( [1], [2]). Early deep-learning systems demonstrated automatic nutrient estimation with a single meal image ( [3], [4], [5]), resulting in ~15-20% mean error. Building upon this foundational work, the field is increasingly leveraging Large Multimodal Models (LMMs), which can reason over different modalities of data like images and text, to advance nutrition analysis. LMMs unify the nutritional analysis task into a single forward pass, which is a key advantage over traditional computer vision pipelines that require separate models. This enhanced accuracy is crucial for clinical applications, as precise dietary assessment supports personalized interventions for managing chronic diseases, weight management, and addressing nutritional deficiencies [6].

However, LMMs for dietary assessment face two primary challenges. First, when cues such as location, meal-time, or individual food items are missing, today's LMMs often hallucinate portion sizes or confuse region-specific dishes [7]–[10]. This is unlike registered dietitians who, in their comprehensive dietary assessments, routinely integrate detailed information on individual food items, precise meal timing, and the broader context of eating occasions [11]. Second, model accuracy is dependent on how the input text-based prompt is phrased, with reasoning modifiers having cut calorie MAE by up to 50% [7], [12], [13]. These findings are often anecdotal, as they draw on fewer than 200 images or on private datasets.

No open benchmark yet pairs the evaluation of dietitian-verified nutrients with location, meal-time, and individual food items, or tests whether modern LMMs can exploit that context under varied prompting. We close this gap by (i) benchmarking eight modern LMMs: four closed-weight (GPT-4o, GPT-4.1, Claude 3 Sonnet, Gemini 2.5 Pro) and four open-weight (DeepSeek Janus-Pro-VL, Gemma-3-IT, Llama-3.2-VI, Qwen 2.5-VL); (ii) iterating through three metadata facets (location, time, food list) with five reasoning input prompt modifiers (chain-of-thought (CoT), multimodal CoT, scale-hint, expert persona, few-shot); and (iii) curating the first publicly available dataset of controlled-feeding meal images with dietitian-verified nutrients and gram-level portions that is aligned with GPS coordinates and second-level timestamps. Automatic capture of this metadata by smartphones makes its integration seamless and reflective of real-world data availability. The dataset was collected with explicit informed consent from all participants. Figure 1 visualizes how location, time, and food-list metadata are fused into a single input prompt for a given model. For the remainder of our study, we will refer to these metadata facets as "contextual metadata".

By systematically integrating contextual metadata and diverse prompting strategies across a wide array of pre-trained LMM models, this study provides insights for practitioners

*B. Coburn and F. Zhu are with Purdue University, West Lafayette, IN 47906 USA (e-mail: coburn6@purdue.edu; zhu0@purdue.edu).

†J. He is with Indiana University, Bloomington, IN 47405 USA (e-mail: jhe2@iu.edu).

‡M. E. Rollo, S. S. Dhaliwal, and D. A. Kerr are with Curtin University, Bentley WA 6102, Australia (e-mail: Megan.Rollo@curtin.edu.au; S.Dhaliwal@curtin.edu.au; D.Kerr@curtin.edu.au).

This study was funded by an Australian Research Council Discovery Project 190101723 entitled "Accuracy and Cost-Effectiveness of Technology-Assisted Dietary Assessment".

§Corresponding author & Project lead.

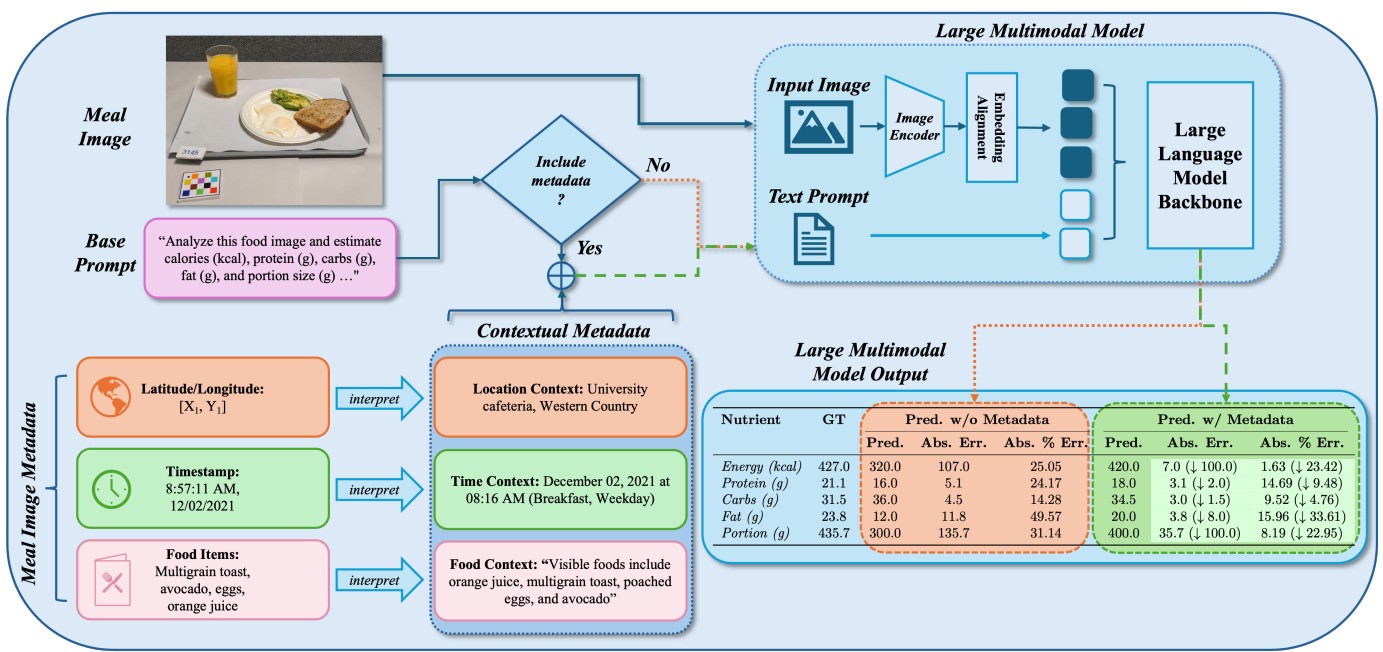

Figure 1: Contextual metadata overview. Location, time, and food context can be combined with the meal photo and the "base prompt" ("Analyze the food image and estimate ..."). This enriched prompt is passed to an LMM to enhance absolute error and absolute percentage error. For this image, energy absolute error and energy absolute percentage error improve by 100 and 23.42 points, respectively. Aggregated results appear in the Results section.

| Nutrient | GT | Pred. w/o Metadata | | | Pred. w/ Metadata | | |
|---|---|---|---|---|---|---|---|
| | | Pred. | Abs. Err. | Abs. % Err. | Pred. | Abs. Err. | Abs. % Err. |
| Energy (kcal) | 427.0 | 320.0 | 107.0 | 25.05 | 420.0 | 7.0 (↓ 100.0) | 1.63 (↓ 23.42) |
| Protein (g) | 21.1 | 16.0 | 5.1 | 24.17 | 18.0 | 3.1 (↓ 2.0) | 14.69 (↓ 9.48) |
| Carbs (g) | 31.5 | 36.0 | 4.5 | 14.28 | 34.5 | 3.0 (↓ 1.5) | 9.52 (↓ 4.76) |
| Fat (g) | 23.8 | 12.0 | 11.8 | 49.57 | 20.0 | 3.8 (↓ 8.0) | 15.96 (↓ 33.61) |
| Portion (g) | 435.7 | 300.0 | 135.7 | 31.14 | 400.0 | 35.7 (↓ 100.0) | 8.19 (↓ 22.95) |

aiming to deploy these models for large-scale, automated dietary monitoring. We illuminate not only the inherent performance of these pretrained models with and without reasoning modifiers for nutrition analysis, but also how different forms of meal context contribute to enhancing LMM nutrition accuracy.

## II. RELATED WORKS

### A. Traditional & Image-Based Nutrition Analysis

Population-scale studies still rely on self-report tools like 24-hour recalls or apps like MyFitnessPal, which are prone to recall bias and systematic error ( [1], [2], [14], [15]). While controlled feeding is the gold standard [16], it is costly. These limitations spurred the development of image-based solutions. Building on Im2Calories' monocular energy-estimation framework [3], later studies leveraged techniques such as 3D reconstruction [17], depth-camera geometry [18], and fusing phone-based RGB-D and IMU modalities to reduce scale error [5]. Multimodal pairing with glucose lowers energy error approximately 15% [19], and DeepFood outputs per-item nutrients in one pass [20]. Modern vision-based solutions jointly localize, classify, and estimate portions [21], rebalance long-tailed logs (Food101-LT, VFN-LT) [22], and employ coarse-to-fine hierarchies to reduce ambiguity [23]. We build on this trajectory by testing whether contextual metadata can further boost LMMs for nutrition estimation.

### B. Large Multimodal Models for Nutrition Analysis

Lo *et al.* provided the first comprehensive analysis of GPT-4V for nutrition estimation and food classification, highlight-ing scale reasoning and hallucination pitfalls [7]. While a foundational contribution, a direct, reproducible comparison with our work is challenging as their study utilized a closed dataset, preventing comparative benchmarking. The Chain-of-Thought prompting approach employed in their study, however, is covered and evaluated within our own evaluation. Existing public datasets for nutrition analysis often lack comprehensive contextual metadata (GPS, timestamp, food item lists) or controlled-feeding ground truth. This limitation hinders direct, reproducible comparisons with current LMMs on contextual metadata. Our work addresses this by providing a standardized benchmark for future evaluations. Kim *et al.* demonstrated that textual descriptors produced by an LMM can be cross-attended with vision transformers to raise Food-101 accuracy [24]. O'Hara *et al.* reported that chain-of-thought and expert-persona prompts halve macronutrient error for simple dishes but still under-estimate complex meals [13].

## III. METHODOLOGY

### A. The ACETADA Dataset

The ACETADA dietary study is a controlled-feeding, randomized crossover trial that involved 152 adults in Perth, Western Australia (55% women; $32 \pm 11$ y, BMI $26 \pm 5$ kg m$^{-2}$) [25], [26]. During three non-consecutive days, participants consumed laboratory-prepared meals (breakfast, lunch, and dinner) that were unobtrusively weighed (to 0.1 g). Meal images were captured using the mFR24 smartphone app [27], [28], which automatically embedded contextual metadata (size reference, timestamp, GPS) into the EXIF. Accredited practic-

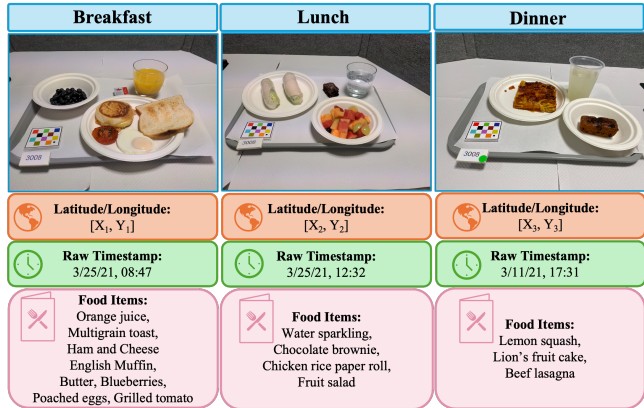

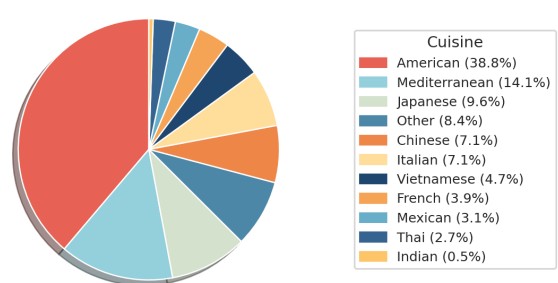

(a) Example images across meals, with corresponding metadata.

(b) Dataset cuisine breakdown—top 10 categories shown; all others grouped as "Other".

Figure 2: **(a)** Example images across meal types alongside metadata. **(b)** Cuisine labels assigned via the method outlined in (Section III-A).

ing dietitians reviewed the images, enumerated visible items, and assigned portion weights and macronutrient profiles. As consumed mass is calculated as served minus leftovers, the "before-meal" image provides the upper bound for estimation.

We used 806 "before-meal" images (36% breakfast, 32% lunch, 32% dinner) for nutrient estimation. Nutrient content varied widely: $83$–$1,968$ kcal (median $622$ kcal) of energy, $2$–$110$ g (median $22$ g) of fat, $3$–$95$ g (median $29$ g) of protein, and $3$–$332$ g (median $71$ g) of carbohydrates, with a median of five items per meal spanning 11 cuisines (predicted by BART [29], [30]). This unique dataset, combining dietitian-verified labels and smartphone images with timestamp and GPS metadata, is unparalleled among existing benchmarks [7], [13], [31], making it highly suitable for realistic nutrition estimation (see Figure 2a for examples). "After-meal" images is also included in the public release for intake studies.

### B. Prompt Engineering with Contextual Metadata

Figure 3 summarizes our prompt-construction approach. A fixed baseline nutrition prompt can be augmented with three metadata facets: gps (location determined from co-ordinates), timestamp (meal type derived from captured time), and food (ground-truth dish names). The prompt can be prepended by a reasoning modifier. The modifiers follow established zero-shot methods: pixel-grounded multimodal CoT (multimodal_cot) [32], scale hinting (scale) [33], and few-shot exemplars (few_shot) [34]. To align with the evaluation methods in [7], we incorporate Chain-of-Thought (CoT) (cot) [35] and an expert-dietitian persona (expert) [12]. The desired prompt is assembled, locks image resolution, and submits identical prompt–image pairs to every model, letting us isolate the effect of each metadata facet and reasoning modifier on nutrition estimation accuracy.

## IV. RESULTS

For our evaluation, we primarily conducted two experiments: one to isolate the standalone benefit of contextual

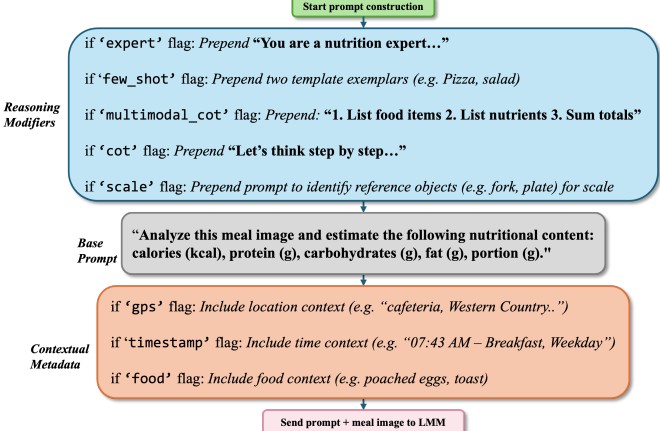

Figure 3: Prompt–construction pipeline. The orange box show metadata facets (gps, timestamp, food); the blue box shows reasoning modifiers (cot, multimodal_cot, scale, fewshot, expert).

metadata and a second to examine its impact in conjunction with reasoning modifiers.

### A. Experimental Setup

All evaluations were conducted on the "before-meal" images of our dataset to establish a baseline regarding nutrition analysis. Each image was paired with a text prompt as shown in Figure 1. Closed-weight models were queried at temperature 0.1 with the most recent checkpoint available, while open-weight models used bfloat16 on NVIDIA H100 GPUs. Default hyperparameters were kept unchanged to ensure any performance difference was attributable to the prompts, not low-level tuning.

### B. Model Suite

Eight modern LMMs were evaluated, where each model takes an RGB meal image and a text prompt as inputs,

and returns a single natural-language response containing the model's estimates for all nutrients (energy, protein, carbohydrates, fat, and portion sizes).

**Closed-weight Models.** We utilize GPT-4o as a high-end baseline model. GPT-4o unifies vision and language, and offers a 128k-token context [36]. Previous GPT-4 Vision models, evaluated by O'Hara et al. and Lo et al. [7], [13], are no longer accessible, making GPT-4o align with modern capabilities and ensure reproducibility. GPT-4.1 extends the same architecture with a 1M-token context [37]. Claude 3.7 Sonnet [38] features a 200k-token window and low hallucination rates. Gemini 2.5 Pro [39] uses a sparse Mixture-of-Experts backbone, supporting up to 1M tokens.

**Open-weight Models.** Among open-weight models, Janus-Pro-VL (7B) [40] is lightweight and designed for efficient vision–language inference. Mid-scale Gemma-3-IT (27B) [41] offers multimodal support with a 128k-token context window. High-end options include LLaMA-3.2-VI (90B) [42] and Qwen 2.5-VL (72B) [43], featuring large context sizes and powerful vision backbones.

### C. Evaluation Protocol and Metrics

For each nutritional attribute $y \in \{\text{kcal}, \text{protein}, \text{carbs}, \text{fat}, \text{portion}\}$ we report the mean-absolute error (MAE) and the mean-absolute percentage error (MAPE):

$$\text{MAE}(y) = \frac{1}{N}\sum_{i=1}^{N}|y_i - \hat{y}_i|, \quad \text{MAPE}(y) = \frac{100}{N}\sum_{i=1}^{N}\left|\frac{y_i - \hat{y}_i}{y_i}\right|,$$

where $N$ is the number of evaluation images, $y_i$ the dietitian ground truth, and $\hat{y}_i$ the model prediction. Section IV shows macro-averaged scores rather than per-model results. For any subset of models $M$ (all models, only open-weight, only closed-weight, or any set of prompting methods), the scheme-level MAE is:

$$\overline{\text{MAE}}_{\text{scheme}} = \frac{1}{|M|}\sum_{m \in M}\frac{1}{|N|}\sum_{i \in \mathcal{I}}|y_i - \hat{y}_i^{(m)}|,$$

with a similar definition for MAPE. These metrics form the basis of all tables and radar plots in Section IV.

### D. Experiment 1: Impact of Metadata on Simple Prompting

We quantified the benefit of contextual metadata by comparing a baseline prompt against variants augmented with all combinations of contextual metadata. Table I reports the best-performing metadata combination for each model. Across the eight models, adding context consistently lowered energy error, with an average decrease in energy MAE of $\sim$ 76 kcal. The most dramatic improvement was for Janus-Pro with the `gps+timestamp` flag, which lowered energy MAE by 246 kcal and energy MAPE by 52 percentage points. Portion size estimates also improved substantially, dropping by an average of about 53g and by as much as 124g for LLaMA-3.2-VI. Protein, carbohydrate, and fat benefitted more modestly, with typical MAE reductions of 2-5g. The only notable adverse

effects were a marginal 0.68g rise in protein MAE for Janus-Pro and a 1 kcal rise in carb MAE for GPT-4o, though all of their other metrics improved.

Performance patterns varied between open-weight and closed-weight models. Closed-weight models showed moderate yet consistent benefits, with Gemini 2.5-Pro lowering every nutrient metric. In contrast, open-weight models recorded the most dramatic improvements. Every "best metadata" combination included either `gps` or `timestamp`, suggesting that location and meal-time cues are the most valuable annotations. Yet, the comprehensive `gps+timestamp+food` combination does not always yield the best performance. For instance, Gemma-3-IT improved the most with only `gps` metadata. This suggests that providing excessive information can sometimes hinder performance, possibly due to overlap with existing internal knowledge or introduced ambiguities. These results highlight model-specific nuances in how LMMs leverage contextual cues, indicating that metadata inclusion consistently improves performance but its compounding effect depends on the model's architecture and pretraining data.

### E. Experiment 2: Impact of Metadata on Reasoning Modifiers

We explored whether contextual metadata complements five widely used reasoning modifiers: *Chain-of-Thought (CoT), Multimodal CoT, Scale-Hint, Few-Shot Exemplars* and *Expert Persona*. For each modifier, we compared a baseline prompt against a metadata-enriched prompt. The most effective combination for every modifier is reported in Table I.

Across all five modifiers, providing contextual metadata consistently lowered prediction error. For instance, the decrease in energy MAE ranged from 21.31 kcal (`few-shot`) to 75.39 kcal (`expert`). The best variant for every modifier included at least one of the `gps` or `timestamp` flags, underscoring the importance of these cues in accurate nutrition estimation.

Similar to Experiment 1, the "best metadata combination" for each reasoning modifier also exhibits variability. This suggests that different reasoning modifiers may benefit from distinct types or subsets of contextual information to achieve better performance. For example, Expert-Persona, designed to direct the LMM to act as a dietitian, showed the best Energy MAE when supplied with all contextual metadata (`gps+timestamp+food`). This aligns intuitively with how human dietitians incorporate all contextual information during their assessments. In contrast, the Multimodal CoT prompt achieved its best results with `gps+timestamp` possibly because the food item list might become redundant or less critical when the model is already directed to identify and list items from the image. These differing patterns underscore that the effect of contextual metadata is not simply additive, but rather synergistic.

### V. DISCUSSION

Incorporating location and meal-time metadata consistently improved nutrition estimates across all eight LMMs, reasoning template, and nutrient metric. Since smartphones automatically

Table I: **Impact of Metadata Integration on LMM Nutritional Estimation Error.** This table presents the change in MAE and MAPE when using the best-performing metadata combination relative to a baseline prompt for each model. (↓) and a green highlight indicate error reduction, (↑) and a red highlight indicate error increase. The best baseline for each metric is highlighted blue and the largest error reduction is bolded for each nutrient. *Legend*: g = `gps`, t = `timestamp`, f = `food`.

| Model | Weight Type | Scheme | Energy (kcal) (↓) MAE (kcal) | (↓) MAPE (%) | Protein (g) (↓) MAE (g) | (↓) MAPE (%) | Carbs (g) MAE (g) | (↓) MAPE (%) | Fat (g) (↓) MAE (g) | (↓) MAPE (%) | Portion (g) MAE (g) | (↓) MAPE (%) |
|---|---|---|---|---|---|---|---|---|---|---|---|---|
| Claude-Sonnet | Closed | Baseline | 181.68 | 23.88 | 15.41 | 36.02 | 25.01 | 37.22 | 7.82 | 41.91 | 329.89 | 42.81 |
| | | w/ (g+t+f) | ↓27.91 | ↓2.56 | ↓4.99 | ↓9.05 | ↓3.82 | ↓2.32 | ↓1.74 | ↓14.45 | ↓55.52 | ↓7.36 |
| GPT-4.1 | Closed | Baseline | 170.02 | 23.29 | 14.33 | 33.00 | 22.90 | 39.34 | 7.58 | 32.21 | 315.83 | 40.62 |
| | | w/ (t+f) | ↓16.83 | ↓2.03 | ↓2.09 | ↓6.06 | ↓2.54 | ↓4.38 | ↓0.22 | ↓1.16 | ↓19.73 | ↓2.53 |
| GPT-4o | Closed | Baseline | 165.77 | 24.48 | 12.67 | 31.15 | 23.37 | 42.08 | 7.53 | 37.67 | 259.96 | 33.37 |
| | | w/ (g+f) | ↓11.61 | ↓1.53 | ↓2.63 | ↓5.85 | ↑1.00 | ↓1.04 | ↓0.21 | ↓3.82 | ↓49.46 | ↓6.79 |
| Gemini2.5-Pro | Closed | Baseline | 211.04 | 34.11 | 12.45 | 39.67 | 28.47 | 50.15 | 10.84 | 53.96 | 188.12 | 25.77 |
| | | w/ (g+t+f) | ↓41.37 | ↓6.28 | ↓2.31 | ↓9.19 | ↓1.80 | ↓3.51 | ↓2.20 | ↓12.37 | ↓11.99 | ↓1.01 |
| Gemma3 | Open | Baseline | 187.54 | 27.67 | 13.72 | 36.49 | 24.51 | 51.15 | 9.47 | 61.96 | 332.00 | 41.61 |
| | | w/ (g) | ↓10.35 | ↑2.49 | ↓2.22 | ↑1.39 | ↓2.30 | ↓1.94 | ↑1.12 | ↑7.00 | ↓67.44 | ↓9.39 |
| Janus-Pro | Open | Baseline | 477.79 | 86.29 | 20.68 | 50.75 | 53.31 | 67.23 | 14.58 | 58.70 | 656.88 | 90.02 |
| | | w/ (g+t) | ↓**246.46** | ↓**52.20** | ↑0.68 | ↓0.64 | ↓2.62 | ↓4.28 | ↓1.70 | ↓9.09 | ↓70.87 | ↓9.35 |
| LLaMA-3.2-VI | Open | Baseline | 496.47 | 66.22 | 38.46 | 117.71 | 45.61 | 66.79 | 45.74 | 200.99 | 483.03 | 62.31 |
| | | w/ (g+t) | ↓193.25 | ↓28.90 | ↓**7.61** | ↓**20.12** | ↓**16.82** | ↓**13.13** | ↓**14.45** | ↓**52.44** | ↓**123.67** | ↓**16.23** |
| Qwen2.5-VL | Open | Baseline | 276.04 | 33.90 | 15.70 | 37.16 | 38.35 | 50.02 | 10.86 | 43.18 | 497.17 | 66.66 |
| | | w/ (t+f) | ↓59.58 | ↓6.20 | ↓2.68 | ↓1.57 | ↓6.10 | ↑0.30 | ↓2.37 | ↓6.52 | ↓23.26 | ↓3.39 |

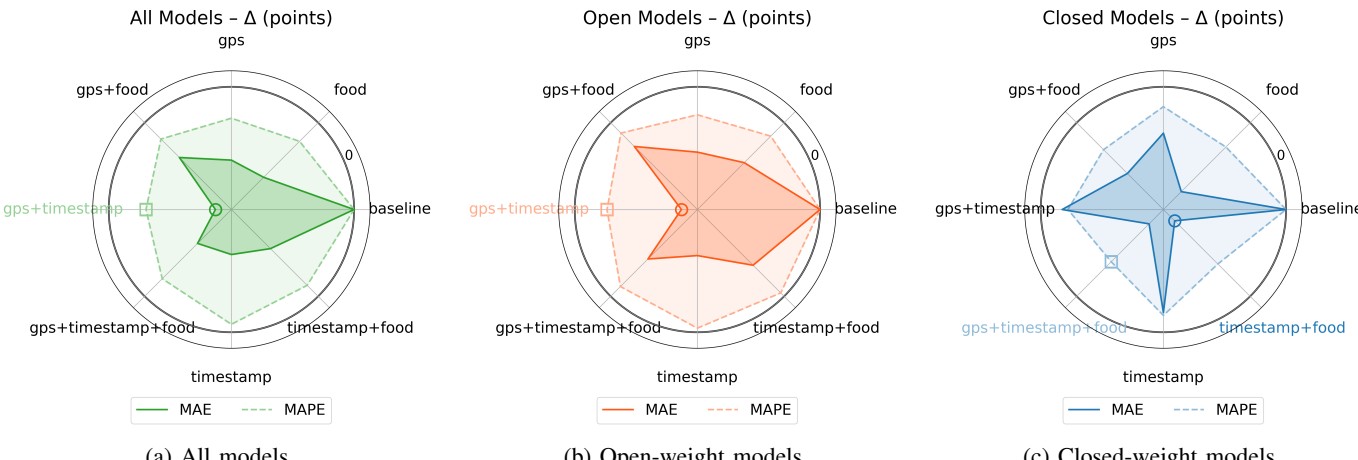

|(a) All models | (b) Open-weight models | (c) Closed-weight models|
|---|---|---|

Figure 4: **Averaged Experiment 1 Results According to Weight-Type.** MAE (solid lines) and MAPE (dashed lines) for various contextual metadata combinations. Each spoke represents error for a given combination. Proximity to the center signifies error reduction relative to the baseline prompt. Colored markers denote the BEST-METADATA configuration for each metric.

capture this information, this integration offers a practical and non-burdensome enhancement immediately deployable in self-monitoring dietary and clinical assessment applications.

The benefits of contextual metadata varied between open-weight and closed-weight models. Open-weight models exhibited the most significant gains, occasionally surpassing proprietary models with better baseline performance. This suggests that proprietary models may have already internalized contextual signals through curation or Reinforcement Learning from Human Feedback (RLHF), leaving less room for improvement. Conversely, open-weight models, despite weaker default priors, proved highly receptive to these cues. This indicates a trade-off: closed models provide strong "off-the-shelf" performance, yet open models can close the performance gap when proper contextual information is supplied at inference.

The performance gains from integrating location and meal-

time data are clinically meaningful. Average reductions of approximately 76 kcal in energy MAE and 53 g in portion size MAE represent substantial improvements in dietary assessment accuracy. These advancements, which require no additional fine-tuning, offer a low-cost path to more precise and personalized nutritional interventions for both individuals and clinicians. While our dataset images originate from a controlled feeding setting, this study serves as a proof of concept that contextual metadata enhances an LMM's ability to estimate nutrition, suggesting its importance for future meal data collection strategies.

## VI. CONCLUSIONS AND FUTURE WORK

This study introduces the first public meal-image dataset pairing dietitian-verified nutrition labels with GPS coordinates, timestamps, and food lists. Leveraging this resource, we eval-

Table II: **Impact of Metadata Integration on Reasoning Modifier Nutritional Estimation Error.** This table presents the change in MAE and MAPE when using the best-performing metadata combination relative to a baseline reasoning modifier, averaged across all models. (↓) and a green highlight indicate error reduction, (↑) and a red highlight indicate error increase. The best baseline for each metric is highlighted blue and the largest error reduction is bolded for each nutrient. *Legend*: `g = gps`, `t = timestamp`, `f = food`.

| Reasoning Mod. | Scheme | Energy (kcal) | | Protein (g) | | Carbs (g) | | Fat (g) | | Portion (g) | |
|---|---|---|---|---|---|---|---|---|---|---|---|
| | | (↓) MAE (kcal) | (↓) MAPE (%) | (↓) MAE (g) | (↓) MAPE (%) | (↓) MAE (g) | (↓) MAPE (%) | (↓) MAE (g) | (↓) MAPE (%) | (↓) MAE (g) | (↓) MAPE (%) |
| Chain-of-Thought | Baseline | 255.00 | 37.07 | 16.26 | 41.08 | 30.39 | 46.23 | 10.15 | 46.67 | 350.20 | 46.18 |
| | w/ (f+t) | ↓51.08 | ↓7.09 | ↓2.26 | ↓**5.14** | ↓**3.68** | ↓3.62 | ↓1.31 | ↓**7.77** | ↓20.92 | ↓1.61 |
| Expert Persona | Baseline | 284.63 | 40.56 | 16.28 | 39.77 | 32.51 | 48.69 | 10.59 | 48.56 | 386.38 | 51.47 |
| | w/ (g+f+t) | ↓**75.39** | ↓**10.38** | ↓2.29 | ↓4.09 | ↓3.44 | ↓0.74 | ↓**1.67** | ↓7.31 | ↓**45.28** | ↓**6.00** |
| Few-Shot Exemplar | Baseline | 240.44 | 33.05 | 15.78 | 38.27 | 31.86 | 47.28 | 10.05 | 46.37 | 381.45 | 50.96 |
| | w/ (g) | ↓21.31 | ↓4.26 | ↓0.72 | ↓2.06 | ↓2.53 | ↓**4.60** | ↓0.68 | ↓5.74 | ↓7.72 | ↓2.15 |
| Multimodal CoT | Baseline | 274.43 | 40.54 | 15.62 | 40.09 | 32.44 | 52.16 | 11.54 | 51.14 | 360.51 | 47.49 |
| | w/ (g+t) | ↓64.46 | ↓7.72 | ↑0.25 | ↓0.64 | ↓1.54 | ↓3.63 | ↓1.11 | ↓5.60 | ↓44.14 | ↓4.41 |
| Scale-Hint | Baseline | 238.79 | 34.44 | 15.68 | 38.46 | 30.74 | 47.57 | 9.91 | 44.81 | 376.27 | 50.54 |
| | w/ (g+f+t) | ↓40.49 | ↓5.67 | ↓**2.42** | ↓2.96 | ↓2.60 | ↑0.48 | ↓1.22 | ↓4.49 | ↓43.64 | ↓5.09 |

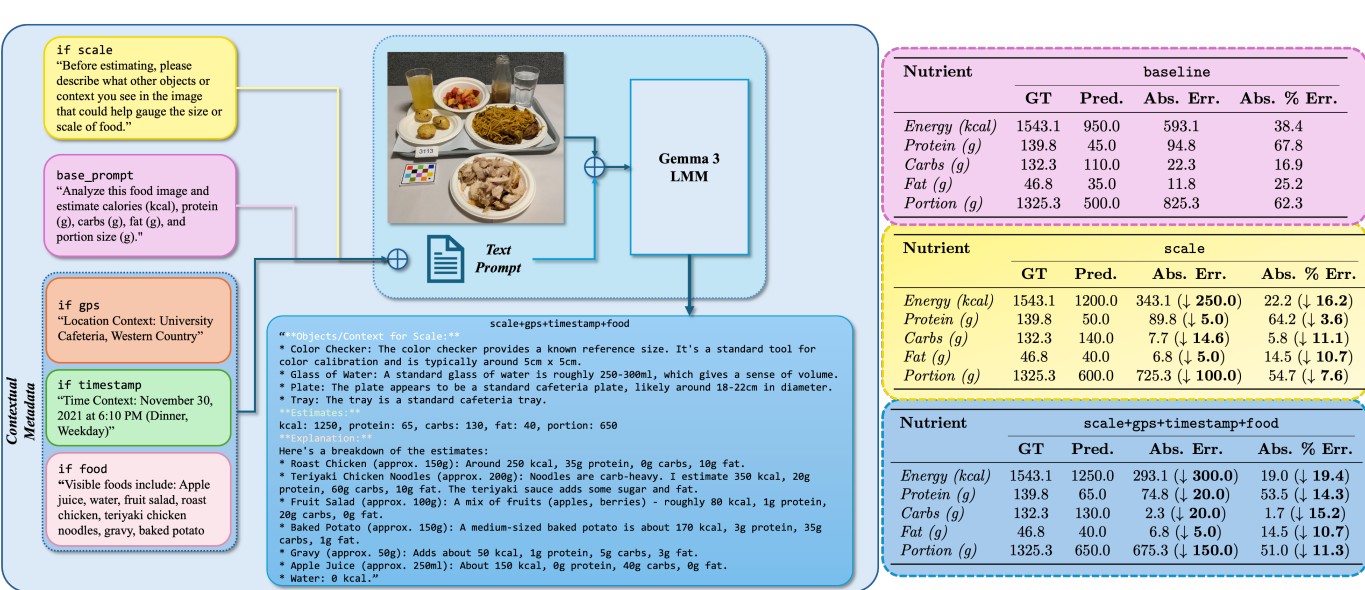

Figure 5: Experiment 2 LMM evaluation example: Gemma 3 with `scale` reasoning modifier. This figure compares baseline (purple), `scale` (yellow), and full metadata (blue) prompt strategies. For this instance, full metadata (blue) improves accuracy over baseline/`scale` for most nutrients (excluding fat).

uated eight LMMs, three metadata facets, and five reasoning modifiers. We found that incorporating interpreted metadata into prompts reduces MAE and MAPE across nutrition metrics, rarely diminishes accuracy, and is particularly effective for open-weight models. This highlights its value for privacy-constrained or on-premise deployments.

Several directions for future work follow. Metadata investigations incorporating behavioral or personal cues could better contextualize portion predictions. Architecturally, future models might benefit from integrating metadata into hidden representations rather than solely relying on text concatenation. Furthermore, a granular error analysis across specific meal types, cuisines, or demographic groups represents an important direction to uncover fine-grained insights into LMM performance. Finally, to evaluate clinical readiness, reliability should be assessed under degraded, real-world conditions.

## DATA AVAILABILITY

The complete ACETADA dataset, including dietitian-verified nutrition labels, meal images, and contextual metadata, is publicly available under the CC BY 4.0 license. The dataset is hosted at https://skynet.ecn.purdue.edu/~coburn6/ACETADA/.

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
