# OpenReview forum: "Evaluating Large Multimodal Models for Nutrition Analysis: A New Benchmark Enriched with Contextual Metadata"
_IEEE.org/EMBS/BHI/2025/Conference — BHI 2025_

### Official Review · Reviewer_587i · 2025-06-30
**Evaluating Large Multimodal Models for Nutrition Analysis: A New Benchmark Enriched with Contextual Metadata**

**Confidence:** 4
**Clarity Of Writing:** fair
**Clinical Significance:** poor
**Methodological Novelty:** fair
**Overall Rating:** 6

**Experiments And Results:**

good

**Questions For The Authors:**

1.	Authors mentioned that the "before-meal" image serves as an upper bound on consumption. However, did you explicitly weigh the served food quantities for each item? If not, how do you control for discrepancies in plating that visually appear larger but have the same quantity?
2.	The scaling prompt appears to anchor portion estimates by providing fixed-size references or interpretive cues. Can you clarify how this prompt was constructed? Was the scaling information automatically inferred from the image (e.g., via object size or marker detection), or was it manually inserted?
3.	What fallback strategies do the authors propose when GPS or timestamps are missing or corrupted in user input?
4.	Could "after" images be used to quantify actual intake or food waste? If so, how do the authors envision this evolving in future work?

**Strengths:**

1.	The dataset is the first to combine dietitian-verified nutritional labels with contextual metadata, and it is intended for public release.
2.	Eight LMMs (open and closed-weight) were rigorously benchmarked across five reasoning strategies
3.	The finding that lightweight metadata (GPS, meal time) can significantly improve estimation accuracy without retraining models is immediately actionable for real-world deployment.

**Summary Of The Paper:**

This paper presents a new benchmark dataset of 806 dietitian-verified meal images with contextual metadata (GPS, time, food items) for evaluating nutrition analysis using Large Multimodal Models (LMMs). Eight LMMs (4 open-weight, 4 closed-weight) are tested with prompt engineering strategies. Results show that incorporating metadata consistently improves prediction accuracy, especially for open-weight models, highlighting the value of context-aware prompting in dietary assessment.

**Weaknesses:**

1.	While the dataset is derived from a controlled-feeding study, its applicability to "in-the-wild" user-generated images (e.g., varying lighting, occlusions) is not explored.
2.	The study assumes metadata is readily available. In practice, GPS and accurate timestamps may be missing or inaccurate for user-submitted images.
3.	Although the paper discusses MAE/MAPE reductions, it stops short of evaluating whether such improvements are clinically meaningful in dietary assessment or intervention settings.

---

### Official Review · Reviewer_VJaC · 2025-07-14
**A sophisitcated and thorough benchmark for nutrition analysis from food images using LLMs, but the clinical/medical motivation remains unclear**

**Confidence:** 4
**Clarity Of Writing:** good
**Clinical Significance:** poor
**Methodological Novelty:** good
**Overall Rating:** 2
**Final Rating:** 3

**Experiments And Results:**

great

**Questions For The Authors:**

My concern with the manuscript primarily lies in the confusion of its clinical significance.

**Strengths:**

- The paper contributes a new public dataset with expert-level annotations.
- The paper provides a solid description of the benchmarking process used to evaluate different LLMs and prompt modifiers.

**Summary Of The Paper:**

Large language models(LLMs) have been increasingly applied to analyze nutrition, but prior work focused on proprietary LLMs and has not explored the influence of integrating contextual metadata such as location, meal-time, and individual food items.
This paper presents an inaugural analysis on the impact of contextual metadata on LLMs' nutrition analysis ability on food images, but conducting a benchmark study on 8 LLMs, exploring 5 reasoning input prompt modifiers, and curating the first public available dataset of controlled-feeding meal images with dietitian-verified nutrients and gram-level portions that is aligned with GPS coordinate and second-level timestamps.

**Weaknesses:**

- The clinical motivation and utility of the proposed method is unclear. While it's understood that cuing LLMs with location, meal-time, and individual food items would minimize its hallucination on portion sizes region-specific dishes. Obtaining these additional contexts, especially location and meal time can be cumbersome and privacy invasive for users. However, the paper does not sufficiently address how such data would be collected reliably and ethically in a real-world setting. It is recommended that the authors discuss practical strategies for capturing these contextual cues while respecting user privacy and minimizing burden, as well as clarify how their inclusion concretely improves model performance in a measurable way.
- On the other hand, the motivation of using LLMs instead of traditional image/CV-based approaches was not clear. Food recognition and nutrition analysis have also been extensively studied in the computer vision community and many of their methods didn't require additional contextual information. The paper should clarify how LLMs provide unique advantages over these existing methods, and whether the proposed approach can be integrated with or complement traditional CV-based techniques.
- In addition to the real-world practicality, training LLMs with this sensitive information could also trigger concerns with bias, such as regional dietary stereotypes, socioeconomic assumptions, or underrepresentation of minority cuisines in the training data. But the paper never discusses whether there could be biases or how these biases could propagate through the system, leading to inaccurate or inequitable dietary recommendations. The authors are encouraged to discuss how their model addresses or mitigates such risks, and whether any fairness audits, bias evaluations, or dataset balancing techniques were employed to ensure generalizability across diverse populations.
- According to Figure 2, the meal images appear highly standardized—clean, well-lit, and consistently organized with reference color sheets and uniform plates. While such protocols improve model training and control experimental variability, they raise concerns about real-world applicability. In clinical or free-living settings, meals are rarely so neatly presented. The paper should clarify how the model's performance might translate to typical user environments where lighting, plating, and food presentation are highly inconsistent. What is the clinical utility if the model requires such rigid conditions to function effectively?
- Despite the extensive experiments, the significance of the reported results is not clear. What’s the main takeaway aside from providing a new dataset? Given that LLMs are already known to perform well with structured prompts, the novelty of applying prompt modifiers and contextual cues (like location or meal type) needs stronger justification. It's unclear whether the observed improvements stem from unique prompt designs or simply reflect the known behavior of LLMs when provided with cleaner, more complete inputs. Clarifying this would help position the contribution beyond dataset curation.

---

### Official Review · Reviewer_e45M · 2025-07-14
**The experimental results are comprehensive but needs more complements for demonstration.**

**Confidence:** 4
**Clarity Of Writing:** good
**Clinical Significance:** good
**Methodological Novelty:** good
**Overall Rating:** 4
**Final Rating:** 7

**Experiments And Results:**

fair

**Questions For The Authors:**

Please see the weakness.

**Strengths:**

This paper is well-writing and the graphs are clear and informative. The comprehensive results are rich in content and reflect the substantial effort put in by the authors.

**Summary Of The Paper:**

In this paper, the authors introduce a meal image dataset slated for public release, pairing dietitian-verified energy, macronutrients, portions, and three metadata facets: GPS coordinates (location), timestamp (meal-time), and food-item lists. They further benchmark 8 state-to-the-art LMMs (4 open-weight, 4 closed-weight), showing the interpreting and adding metadata to the prompt consistently reduces mean-absolute and mean-absolute-percentage errors relative to baseline prompting.

**Weaknesses:**

But there are still some issues:

(1) Lack Comparison with SOTA methods. According to Section Related Works, there are not many researches about LMM for nutrition analysis. Reference [4] should be the most relevant paper to the authors’, however, the authors don’t compare their proposed method with Reference [4], which makes it not convincing enough. Moreover, for better verification, the authors should search more up-to-date papers for comparison.

(2) Experimental Results are kind of subjective and tricky. According to Table I, the best performing metadata combination varies in different models. For instance, in terms of Gemma3, Baseline w/ GPS performs best, but the authors don’t fully explain the reasons, why GPS + timestamp + food performs inferior to only GPS, since GPS + timestamp + food theoretically provide more prior information to LMMs compared to only GPS. And this option also applies to other models, which means the authors should clearly explain all the results not with “GPS + timestamp + food” for better persuasion.

(3) The results of Table II has the same concern, I would recommend the authors provide the results of all different meta combination and fully explain them.

[4] Dietary Assessment with Multimodal ChatGPT: A Systematic Analysis

---

### Official Review · Reviewer_MogN · 2025-07-18
**Timely application of LLMs in a context clinicians and patients would love to see in the future for nutrition tracking..**

**Confidence:** 3
**Clarity Of Writing:** great
**Clinical Significance:** great
**Methodological Novelty:** good
**Overall Rating:** 7

**Experiments And Results:**

great

**Questions For The Authors:**

None.

**Strengths:**

	The dataset availability greatly bolsters the significance of this paper.
	The structured search of 8 models with various metadata and reasoning modifiers provides a good berth of search for how this data impacts models.
	The comparisons of open-weight and closed-weight models provides an intriguing comparison in the discussion about how the open models are better able to adapt to additional metadata
	Figure 3 provides great explanation of what the reasoning modifiers are and the metadata context added

**Summary Of The Paper:**

This paper analyzes the impact of imparting additional metadata context and using various reasoning modifiers to the input into 8 LLMs focused on the task of nutritional estimation of meals based on a provided photo. The paper presents part of a recent dataset collected of 806 meal images from 152 participants, with each meal being estimated and labeled by trained nutritionists.  The models showed improvement with the various combination of added context, but always improved over the baseline estimation from the LLM.  Proposed future application are the easily deployed augmentations to LLM based nutrition analysis from mobile image and analyzing additional metadata cues to pass to the models.

**Weaknesses:**

	While the tables are beneficially color coded, the text is small and difficult to analyze when printed, especially in gray scale.  Restructuring the table to enlarge the font would make the tables easier to read and reference.
	While keeping the summarized and averaged model results helps the paper to focus on key trends, adding an appendix of results could help future researchers who may want to reference the work or a specific combination of metadata/reasoning modifiers which are not directly shown in the summarized Results.

---

### Official Review · Reviewer_s7rW · 2025-07-18
**Review for Evaluating Large Multimodal Models for Nutrition Analysis: A New Benchmark Enriched with Contextual Metadata**

**Confidence:** 4
**Clarity Of Writing:** great
**Clinical Significance:** great
**Methodological Novelty:** good
**Overall Rating:** 7

**Experiments And Results:**

excellent

**Questions For The Authors:**

How consistent are the gains from metadata integration across different meal types (breakfast vs. dinner) or cuisines? A more granular error analysis might reveal whether certain contexts are more helpful than others. If metadata improves some meal types more than others, this could inform future modeling or deployment.

Have the authors considered training a lightweight model specifically tuned to use contextual metadata, rather than relying solely on prompt concatenation? While prompt-only methods are easy to deploy, architectural tuning might yield further gains, especially for open models.

**Strengths:**

The paper evaluates LMMs for structured, high-stakes tasks in nutrition analysis and provides a curated benchmark dataset with contextual metadata and a large-scale comparison across models and prompting strategies. Both contributions are novel and useful for the research community. The experimental design is thorough, isolating the effects of metadata and reasoning modifiers in a controlled manner. The analysis is detailed, with clear results showing both overall trends and model-specific behaviors. The public release of the dataset further enhances the paper’s value as an enabling resource. The paper is clearly written and well-structured, making its findings accessible and reproducible.

**Summary Of The Paper:**

This paper introduces a new benchmark for evaluating large multimodal models (LMMs) in the task of image-based nutrition analysis. The authors curate a controlled-feeding dataset of 806 “before-meal” images, each annotated with dietitian-verified energy, macronutrient, and portion labels, as well as three forms of contextual metadata: GPS location, timestamp, and a list of food items. They evaluate eight recent LMMs (four closed-weight and four open-weight) under different prompting schemes, systematically varying the inclusion of metadata and reasoning modifiers (e.g., Chain-of-Thought, Multimodal CoT, Scale-Hint, Expert Persona, and Few-Shot examples). Results show that contextual metadata consistently improves prediction accuracy across models, and that metadata-enriched prompts amplify the benefits of reasoning strategies. Open-weight models, in particular, show substantial gains when supplied with contextual metadata, in some cases outperforming closed-weight systems. The dataset and code are intended for public release.

**Weaknesses:**

One limitation is that the study only uses “before-meal” images, which represent an upper bound of consumption and may not reflect actual intake. While this is a defensible design choice for nutrient estimation, it may limit applicability to intake estimation tasks.